# Rheological Characteristics of Wheat Dough Containing Powdered Hazelnuts or Walnuts Oil Cakes

**DOI:** 10.3390/foods13010140

**Published:** 2023-12-30

**Authors:** Karolina Pycia, Lesław Juszczak

**Affiliations:** 1Department of Food Technology and Human Nutrition, Institute of Food Technology, College of Natural Science, University of Rzeszow, Zelwerowicza Street 4, 35-601 Rzeszow, Poland; 2Department of Food Analysis and Evaluation of Food Quality, University of Agriculture in Krakow, Balicka Street 122, 30-149 Krakow, Poland; rrjuszcz@cyf-kr.edu.pl

**Keywords:** dough, nut oil cakes, farinograph, water absorption, extensograph, rheological properties

## Abstract

This study assessed edible oil industry byproducts, oil cakes (OC) based on hazelnuts and walnuts (HOC, WOC), to replace wheat flour dough (WD) based on farinograph and extensograph parameters and rheological measurements. The research hypothesis of this work is that replacing part of wheat flour with ground nut oil cakes modifies the rheological characteristics of the dough. WF was replaced at the level of 5%, 10% and 15%. It was shown that use of OC in flour mixtures at various levels significantly influenced the rheological properties of the dough. The water absorption of wheat flour and oil cake mixtures was higher than that of the control sample, and the average value of this indicator was 53.4%. The control sample had the lowest dough development time (DDT), and the presence of HOC or WOC in the system resulted in a significant increase in this parameter (*p* < 0.05). The average DDT of WDHOC cakes was 4.7 min and was lower compared to WDWOC which was 5.9 min. The WDWOC10% and WDWOC15% samples were characterized by the highest dough stability value and the lowest degree of softening (*p* < 0.05). The presence of OC in the flour mixtures increased the values of the storage and loss moduli, which was reflected in the K′ and K″ values. The values of these parameters also increased as the level of OC addition increased. WDHOC doughs were characterized by higher values of the K′ and K″ parameters compared to WDWOC. The results of the creep and recovery test showed that the dough with the addition of nut OC was less susceptible to deformation compared to the control dough (*p* < 0.05). The resistance to deformation increased with the increasing share of HOC or WOC in the mixtures. The average value of viscoelastic compliance (J_1_) of this parameter for WDHOC dough was on average 1.8 × 10^−4^ Pa^−1^, and for WDWOC 2.0 × 10^−4^ Pa^−1^. Nut oil cakes are an interesting technological addition to the dough. Their use may have a positive impact on the characteristics of the finished product and expand their application possibilities in the food industry. This is because the dough with the addition of nut cakes was more stiff and less flexible and susceptible to deformation.

## 1. Introduction

Bakery products are an important element of the daily diet of the vast majority of consumers. Within this food category, consumers most often choose bread. The basic raw material for its production is flour, produced by grinding grain. The degree of the grain milling determines the technological properties and nutritional value of the flour and thus the dough obtained from it. Strong flours, on the basis of which consumers most often prefer bread, are low in important ingredients like minerals, fiber and bioactive substances. During the milling process, the mentioned substances are lost along with the bran. Therefore, there is a need to enrich bread made from light flour with substances that are naturally rich in these compounds and at the same time cheap. Hence, in recent years, there has been an increasing interest among scientists and food producers in the potential possibilities of using by-products from the food industry for food-improving fortifications. Bread plays an excellent role as a carrier in this process. However, the properties of the final product depend on the rheological properties of the dough, which is a complex material in terms of composition and structure. The dough is made by combining water and flour, the chemical composition of which, as well as the type and amount of other technological ingredients, are decisive for its rheological properties [1]. These properties generally reflect the type and quantity of recipe ingredients or technology modifications. This is particularly visible in the analysis of viscoelastic properties, which affect the quality of the resulting bread, mainly the volume of the loaves and the texture of the crumb [2]. In addition, there are other reasons why the rheological properties of dough are important. Firstly, because they first determine the behavior of the dough during mechanical processing, such as kneading, dividing, rounding and shaping. Secondly, rheological properties influence the quality of the finished loaf of bread [3,4]. In the case of dough based on wheat flour, the rheological properties largely depend on the structure of gluten, formed by combining gliadin and glutenin in an appropriate ratio, and on the type of recipe ingredients and their mutual interactions. An interesting recipe ingredient that can be used to enrich bread is hazelnut or walnut oil cake produced as a by-product in the cold pressing of oil. This valuable by-product is a cheap and rich resource of dietary fiber, mineral ingredients and a number of biologically active substances that should return to food and thus nourish the human body [5,6]. In previous work, Pycia and Juszczak [7] found the effect of hazelnuts and walnuts on the farinographic, extensographic and rheological characteristics of wheat dough. However, there is no information regarding the properties of wheat dough containing by-products from the fat pressing process. 

Therefore, the subject of this work was to evaluate the impact of substituted wheat flour with powdered hazelnut or walnut oil cakes at various levels on selected rheological properties of wheat dough determined by means of a farinograph, extensograph and oscillatory rheometer.

## 2. Materials and Methods

### 2.1. Materials

In the present research, a dough based on wheat flour (13.3% protein content, 0.9% fat content, 0.7% ash content, 14.0% moisture content) type 650 (Złote Pola, Poland) and its mixtures with hazelnut-based oil cake (HOC) or walnut-based oil cake (WOC) were examined [8]. HOC (36.4 g/100 g d.m. protein content, 22.8 g/100 g d.m. fat content, 5.3 g/100 g d.m. ash content, 7.5% moisture content) and WOC (42.7 g/100 g d.m. protein content, 22.1 g/100 g d.m. fat content, 5.6 g/100 g d.m. ash content, 5.6% moisture content) (Warmińska Manufaktura, Jeziorany, Poland) were produced in the process of cold pressing oil. In previous work, the content of minerals, fat and fiber in flour and in mixtures with HOC and WOC were determined [8]. The dough was made on the basis of wheat flour mixtures, which were replaced with hazelnut oil cakes (WDHOC) or with walnut oil cakes (WDWOC) in the amount of 5%, 10% and 15%. In the tested mixtures containing HOC, the moisture content, fat, protein, crude fiber and ash content (d.m.) were, respectively, for the 5% supplementation level: 14.1%, 1.7%, 14.6%, 7.4% and 0.7%; for the 10% level: 14.0%, 2.6%, 15.8%, 9.3% and 1.1%; and for the 15% level: 13.5%, 4.3%, 17.1%, 12.0% and 1.3%. In turn, in mixtures containing WOC, the moisture, fat, protein, crude fiber and ash contents (d.m.) were 13.8%, 1.6%, 14.9%, 8.3% and 0.7% for the supplementation level of 5%; for the 10% level: 13.6%, 3.1%, 16.4%, 8.2% and 0.8%; and for the 15% level: 13.5%, 4.5%, 18.2%, 12.9% and 1.2% [8]. The control sample was a dough based on wheat flour without the tested additives. Moisture content [9] was determined in the obtained systems.

### 2.2. Methods

#### 2.2.1. Farinographic and Extensographic Analysis

The farinographic properties of all samples were determined in a farinograph (Farinograph-E, Brabender, Duisburg, Germany) [7,10]. The time of analysis was 15 min. The parameters evaluated were flour water absorption (WA) [%], dough development time (DDT) [min], dough stability [min], degree of softening [BU] and farinographic number. In addition, the extensographic characteristics were analyzed in a Brabender extensograph (Extensograph-E, Brabender, Duisburg, Germany) [7,11]. In this study, dough energy [cm^2^], resistance to extension [BU], extensibility [mm], maximum [BU] and ratio number were determined. 

#### 2.2.2. Rheological Measurements

All rheological analysis was carried out using a MARS II oscillatory rheometer (Thermo Fisher Scientific, Waltham, MA, USA) equipped with a system of parallel corrugated plates (diameter 35 mm, gap size 1 mm), at 25 °C [7]. The dough samples were obtained by thoroughly mixing the wheat flour or its mixtures with nut oil cakes with the amount of water determined by farinographic analysis. Fresh dough was prepared for each analysis. Before measurements, the dough sample was placed in the measuring system of the rheometer and left for 3 min for stress relaxation and to stabilize the temperature.

##### Determination of Yield Stress of Dough

The yield stress of tested dough was determined in the shear stress range of 10–10,000 Pa and the frequency of 1 Hz. The yield stress was determined at the point of intersection of the storage module (G′) and the loss module (G″). The assay was performed in three replicates. 

##### Frequency Sweep Test

The results of the frequency sweep test (mechanical spectra) were obtained in the range of linear viscoelasticity at a constant strain amplitude of 0.1% in the angular frequency range of 1–100 rad/s [7]. The experimental data were described by the power low models:(1)$$G^{\prime }\left(\omega \right)=K’\cdot \omega^{n^{\prime }}$$
(2)$$G^{\prime \prime }\left(\omega \right)=K^{\prime \prime }\cdot \omega^{n^{\prime \prime }}$$
where *G*′—storage modulus (Pa), *G*″—loss modulus (Pa), ω—angular frequency (rad/s), *K′*, *K″*, *n′*, *n″*—experimental constants. 

Based on the values of the dynamic moduli, the tangent of phase shift angle (tan *δ* = *G*″/*G*′) was calculated and its dependence on the angular frequency (*ω*) was plotted for each of the systems.

##### Creep and Recovery Test

The creep and recovery test was carried out with constant creep strain τ_0_ = 2 Pa for 120 s. The recovery test was continued for 240 s [7]. Experimental data were described using the Burgers model:(3)$$J(t)=J_{0}+\frac{t}{\eta_{0}}+J_{1}\cdot (1-\exp^{-t/\lambda_{ret}})$$
(4)$$J(t)=\frac{t_{1}}{\eta_{0}}-J_{1}\cdot (1-\exp^{t_{1}/\lambda_{ret}})\cdot \exp^{-t/\lambda_{ret}}\;$$
where *J*—compliance (Pa^−1^), *J*_0_—immediate compliance (Pa^−1^), *J*_1_—viscoelastic compliance (Pa^−1^), *η*_0_—zero shear viscosity (Pa∙s), *λ_ret_*—retardation time (s), *t*_1_—time after which the stress is removed (s).

#### 2.2.3. Statistical Analysis

The statistical analyses were performed using the Statistica 13.3 program (StatSoft, Tulsa, OK, USA) [8]. Analyses were carried out in at least three repetitions. In order to assess the significance of differences between mean values, two-way analysis of variance and Duncan’s post hoc test were used at a significance level of 0.05. Moreover, the values of linear correlation coefficients were calculated between the selected parameters and their significance was assessed at the 0.05 level. 

## 3. Results and Discussion

### 3.1. Farinograph Characteristics

Table 1 shows the values of flour water absorption and rheological parameters of the tested doughs, determined using a Brabender farinograph. The statistical analysis indicated a significant effect of all tested factors on the values of the obtained rheological parameters of the dough. It was shown that the water absorption capacity of flour (WA) depended significantly only on the type of nut oil cake (*p* < 0.05). The impact of the remaining tested factors (the level of replacing flour with nut oil cakes and the interactions of both factors) turned out to be statistically insignificant (*p* > 0.05). The average WA of wheat flour was 52.3% and was the lowest among all tested samples. However, mixtures containing OC showed higher WA than the control. In the case of mixtures with OC, it was found that those containing WOC had slightly higher water absorption compared to systems containing HOC. The WA of wheat flour determined in the study is slightly lower compared to the results given by other authors. Stojceska and Butler [12] claimed that the WA of flour derived from 24 wheat varieties ranged from 58.8% to 60.6%, with the protein content ranging from 11.4% to 11.9%. However, Sarker et al. [13] claimed that the WA of durum wheat flour is very high and is approximately 72% (protein content 13.2%), and Schmiele et al. [14] determined the WA of wheat flour at an average level of 63.2%. Meanwhile, Pycia and Juszczak [7] found that the WA of wheat flour was also higher than that reported in the work, but lower than that reported by the cited authors, and amounted to 54.9%. In the case of wheat flour, the main components responsible for water absorption are gluten proteins, which can absorb water in an amount as much as three times their weight [14,15] and starch, i.e., a biopolymer with good hydration abilities. In the case of the analyzed flours with the addition of nut oil cakes, a slight increase in their WA was observed. This may probably be due to the increasing share of fiber (cellulose, hemicellulose) due to the addition of oil cakes (OC), which could have slightly increased the WA of the system. Pycia and Juszczak, in their earlier work [8], indicated that the content of crude fiber in flour mixtures containing HOC or WOC increased with the increasing share of oil cakes. This is consistent with the observations of Schmiele et al. [14], who observed an increase in the WA of wheat flour as the share of wheat bran and whole grain flour in it increased. Similar observations were made by Gómez et al. [16] and Wang et al. [17] who found that bran addition affected an increase in the WA of dough. An increase in WA of dough under the influence of extruded bran was also observed by Li et al. [18]. According to other authors, dough that is enriched with fiber containing cellulose and hemicellulose is usually characterized by greater stability, and such flour has significantly higher water absorption [19,20]. Adamczyk et al. [19] observed a varied effect of fiber preparations on the WA of the tested rye flour. As the cited authors demonstrated, replacing part of the flour mixture with buckwheat and beetroot fiber significantly reduced the water absorption of the flour. However, in flour mixtures with the addition of linseed fiber, a significant increase in the value of this parameter was observed. In turn, Pycia and Juszczak [7], in systems in which part of the flour was replaced with ground hazelnuts or walnuts, observed a significant decrease in WA, which was greater the more ground nuts were in the systems. The cited authors explained this by decreasing levels of gluten proteins and increasing levels of fat. Water absorption of flour is an essential technological parameter of the dough produced. The volume of water absorbed by the flour when forming the dough is crucial for its rheological and technological properties, but also affects the mass of the dough and its efficiency [1]. Low flour WA results in low dough yield and low bread yield. Moreover, the knowledge of the flour’s WA makes it possible to properly select the amount of water necessary to produce the dough. Too little water in the recipe leads to a stiff, hard dough, which hinders the fermentation process and adversely affects the volume of baked loaves [14,21]. According to Zhang et al. [22], water absorption of flour depends on the starch content, the size of its grains, the degree of starch damage, protein content and fiber content. As part of the farinographic analysis, the dough development time (DDT) was also determined. Farinographic parameters such as dough development time (DDT) and stability value (ST) provide information about the strength of the flour. However, higher values of these parameters correlate with a stronger dough [18]. It was shown that the control sample had the lowest value of this parameter (3.5 min), and the presence of HOC or WOC resulted in a significant increase in DDT (*p* < 0.05). It was found that the average DDT of WDHOC dough was 4.7 min and was lower compared to the DDT of WDWOC dough, which was 5.9 min. The increase in the development time of doughs containing nut oil cake may result from a change in the chemical composition of the dough. The addition of OC at the expense of flour causes a decrease in the content of gluten proteins in the system and thus a dilution of the gluten network, and here a decrease in DDT should be expected. However, increasing fiber levels [8] in OC doughs resulted in increased DDT. In previous work, Pycia and Juszczak [8] indicated that the higher content of fiber was in systems of wheat flour with a 15% level of HOC/WOC. Therefore, fibers probably absorb water slower than gluten proteins, hence the higher DDT value. This agrees with the statements of other researchers [14]. These researchers observed an increase in DDT in dough based on wheat flour, which was replaced with wheat bran in a 60:40 ratio. Zhang et al. [22] observed an increase in the development time of the dough with pulse flour additions. Adamczyk et al. [19] observed different effects of different fiber preparations on the development time of rye–wheat dough. As the cited authors showed, the effect of flax fiber was insignificant, and the addition of buckwheat fiber contributed to the reduction of the DDT value. In turn, beet fiber resulted in an increase in the value of this parameter. Li et al. [18], discussing the farinographic properties of dough with the addition of bran, including extruded bran, noticed that this ingredient extends the dough development time in the case of low-gluten dough, which reflects the increase in the time needed to obtain a dough with optimal consistency. However, as the gluten content in the tested flour increased due to the addition of bran, a decrease in the value of this parameter was found. Jakubczyk and Haber [23] and Achremowicz et al. [24] used DDT to determine the quantity and quality of gluten in flour and its water absorption capacity. Flours containing weak gluten bind water quickly and therefore have a short DDT. However, those rich in strong gluten bind water very slowly and therefore their DDT lasts longer. In the analyzed samples, the stability of the dough, i.e., its stability time, was also determined. This parameter is given in minutes and is calculated from the end of dough development until the dough consistency begins to drop below 500 BU [24]. The stability of the dough depends mainly on the quantity and quality of gluten contained in the flour.

It was found that dough stability ranged from 5.5 ± 0.4 min (WDHOC15%) to 13.7 ± 0.3 min (WDWOC15%). The average stability of the WDHOC dough was higher compared to the control sample and WDWOC by 3% and 98%, respectively. Therefore, the addition of OC, especially WOC, had an effect on the dough because it increased its stability. This may be due to the low fat and higher fiber content of the dough [8]. The stability of wheat dough determined in the work is much lower compared to the values reported by other authors [7]. These researchers indicate that the stability of wheat dough was 11.6 min, and this parameter decreased due to the presence of hazelnuts or walnuts in the dough (*p* < 0.05). Schmiele et al. [14] noticed that wheat dough containing wheat bran and whole grain flour differed from wheat dough in terms of stability. The authors showed that only the dough containing 40% wheat bran showed stability similar to the control sample. According to the authors, the dough was stiff and its stability was due to the high content of fiber. Li et al. [18] showed that the increasing share of bran in the system increases the durability of the dough—it was longer in the case of flour with low gluten content. In turn, in the case of flour with a higher gluten content, the durability of the dough decreased as the amount of bran in the flour increased. According to the cited authors, the extension of dough development time and dough stability in low-gluten flour may be caused by, among others, the influence of bran on the gluten formation process as a result of competition for water. This prolonged the dough rising process but improved its stability. Moreover, the improvement in dough stability after the addition of bran is probably due to the increased stiffness of the dough. In the presented tests, WDWOC dough was more stiff compared to the control and WDHOC samples. This translates into the behavior of the dough during its production. The stability time of the dough indicates the strength of the flour. Replacing wheat flour with an alternative flour generally reduced the strength of the dough. The reason for this is the dilution of gluten, which adversely affects the development of the gluten network in the dough [25]. The sum of the development time and stability indicates the resistance of the dough to mixing. The mixing tolerance index was used to assess the softening degree of the dough [18]. The higher this value, the longer the dough should be mixed. This means that the WDWOC15% dough should be mixed for the longest time. The technological properties of the dough are also determined by the degree of dough softening determined as a result of farinographic analysis. The softening of the dough results from the weakening of its structure, mainly the gluten network, which can be observed by reducing the resistance of the farinograph mixers. It was found that the softening degree of the control dough was 64 BU and was higher by 18% and 70%, respectively, compared to WDHOC and WDWOC. Therefore, WDWOC15% dough turned out to be the stiffest, which may probably be related to the higher fiber content. Pycia and Juszczak [7] showed an increase in the degree of softening of wheat dough containing ground hazelnuts or walnuts. In the presented research, doughs containing nut oil cakes had higher farinographic numbers compared to the control sample. The WDWOC15% sample had the highest value of this parameter. Guardianelli et al. [26] showed that farinographic parameters such as WA, DDT and dough stability decreased significantly with the addition of a by-product based on pistachio seeds. However, an increasing degree of softening was observed. Moreover, Tarek-Tilistyák et al. [27] examined the influence of oil cake obtained from naked pumpkin seeds, sunflower seeds, yellow flax and walnut on the rheological properties of wheat dough. The cited authors found an increase in water absorption of flour and dough due to the extension of dough development time and a decrease in dough elasticity. This is consistent with the observations made in this work. The reason for this is probably the increased amount of fiber and protein in the dough. In turn, Shongwe et al. [28] found a significant effect of the addition of peanut flour in amounts ranging from 0% to 20% on the rheology of dough and the physicochemical and sensory properties of bread made from wheat–nut flour. According to the cited authors, the water absorption of flour, DDT and dough stability increased, respectively, from 57.7% to 60.1%, from 2.5 min to 6 min and from 5.5 min to 8.5 min in the case of white bread, along with an increasing share of nut flour ranging from 0% to 20%. In turn, the cited authors found an increase from 62.7% to 64.9%, from 6 min to 7 min and from 8.5 min to 13 min in the case of WA, DDT and dark dough stability. In turn, Hussein et al. [29] found a significant increase in parameters such as water absorption, DDT, mixing tolerance index and dough weakening due to the addition of sprouted nut flour and milk permeate with probiotic bacteria.

### 3.2. Extensograph Characteristics

For the full technological characterization of the dough, in addition to the farinographic analysis, an extensograph was also performed. In the extensographic test, the dough is subjected to shear and uniaxial tension [30]. The results of extensographic tests after 30, 60 and 90 min of fermentation are presented in Table 2 and Table 3. The determined results allow for an approximate assessment of the viscoelastic properties of the dough and enable conclusions regarding the behavior of the dough during the actual technological process [24]. A thorough and comprehensive analysis of the viscous and elastic properties of the dough is possible using an oscillatory rheometer. The results of such tests are presented in Section 3.3. As part of the extensographic analysis, the energy of the dough [cm^2^], i.e., the area of the field under the curve plotted during the analysis, was determined. According to Achremowicz et al. [24], dough intended for baking bread should have high energy, ranging from 90 to 120 cm^2^. It was found that the dough based on wheat flour had energy close to the given range, which proves its good suitability for baking bread. However, the addition of OC had different effects on the energy of the obtained wheat dough, and the changes observed were often irregular. The change in dough energy was particularly visible with the addition of WOC, where the value of this parameter generally increased with the increasing share of this additive (*p* < 0.05). The average energy value of the dough after 30, 60 and 90 min of fermentation was higher by 18, 40 and 45 cm^2^, respectively. It was shown that only in the case of the control dough, the lifting energy of the dough decreased as the fermentation time progressed. In the case of the remaining oil cakes, an ambiguous trend was observed. In the case of WDWOC5% and 10%, an increase in the lifting energy of the dough was observed, which is probably related to the interactions of the ingredients of the oil cakes (fiber, minerals) and the dilution of gluten proteins. Therefore, to loosen such a dough, more force will be needed, i.e., a larger volume of CO_2_. The tensile strength of the tested doughs was also determined after 30, 60 and 90 min of fermentation. It has been shown that, as a rule, the presence of nut OCs in wheat doughs increased their tensile strength. The statistical analysis showed the effect of all tested factors on this parameter measured after 30, 60 and 90 min of fermentation. It was found that the average value of this parameter in the case of WDHOC/WDWOC after 30, 60 and 90 min of fermentation was 490/451 BU, 551/500 BU, 578/551 BU, respectively. Therefore, as the fermentation time progressed, the tensile strength increased in doughs containing nut oil cakes. WDHOC were characterized by greater tensile resistance compared to WDWOC. This behavior of the dough can probably be explained by the increasing share of fiber in the dough and the decreasing amount of gluten. The increasing tensile strength of dough with an increasing share of wheat bran was also demonstrated by Schmiele et al. [14]. The dough tested by these authors was also characterized by increasing tensile strength as fermentation progressed. The cited authors noted that the dough containing 40% wheat bran after 135 min of fermentation recorded a decrease in the value of this indicator compared to dough containing a smaller share of wheat bran. These authors explained a large amount of fiber in a 60:40 ratio causes dilution of gluten, which makes it difficult to create a gluten network due to the resistance of the dough. In turn, Pycia et al. [7], in their studies on the extensographic properties of wheat dough with ground hazelnuts or walnuts, did not observe a significant impact of the presence of nuts on the tensile strength of the dough. The extensibility of all tested doughs decreased as fermentation progressed, which agrees with the statements of other researchers [7,14]. This research showed that the average extensibility of WDHOC/WDWOC doughs after 30, 60 and 90 min of fermentation was 114/144 BU, 108/150 BU, 102/146 BU, respectively. In the case of WDHOC, a clear decrease in dough extensibility was observed with an increase in the share of oil cakes in the system. Schmiele et al. [14] made a similar observation in examining the extensibility of dough containing different bran content. The research also noted that WDWOC doughs were also characterized by greater extensibility compared to WDHOC. Moreover, all WDHOC doughs were characterized by lower extensibility compared to the control and WDWOC samples. Pycia and Juszczak [7] claimed that the extensibility of dough with walnuts was greater than that of wheat dough without additives and the indicator increased with the increasing share of nuts. This behavior of the dough may result from the interaction of many factors, such as a decrease in gluten content, an increase in fiber and fat content or the size of oil cake particles. Quality parameters and the amount of gluten influence the dough’s tensile strength and dough extensibility [31]. Various substances influence the rheological properties of the dough. Dokic et al. [31], examining the effect of replacing wheat flour with chestnut flour, noticed that the gradual replacement of wheat flour with chestnut flour resulted in an increase in the hardness and stiffness of the dough and a decrease in its elasticity. Therefore, the lower gluten content in the dough containing chestnut flour resulted in a reduced degree of gluten binding. As a result of such substitution, the sucrose content in the system increased, which resulted in an intensified interaction of sucrose with starch, which resulted in a crumbly consistency of the dough. According to the cited authors, optimal rheological properties of the dough were obtained with a chestnut flour content of 20%. 

The extensographic test also determined the maximum tensile strength. The conducted two-factor analysis of variance showed the influence of all tested factors on the maximum value (Table 3). The only exception is the type of oil cake, which did not have a statistically significant effect on the maximum dough after 30 and 60 min of fermentation (*p* > 0.05). The conducted research showed that the average maximum value of WDHOC/WDWOC doughs after 30, 60 and 90 min of fermentation was 518/506 BU, 574/596 BU and 596/673 BU, respectively.

### 3.3. Yield Stress and Viscoelastics Properties of Doughs

Wheat dough and doughs with HOC or WOC were subjected to analysis to determine the yield stress. The yield stress was determined experimentally on a figure of the dependence of the storage modulus and the loss modulus on shear stress, at the point of intersection of G′ and G″ (Figure 1). 

The yield stress is a rheological parameter describing the properties of non-Newtonian fluids. The yield stress is the smallest limit shear stress necessary to induce flow. Therefore, below the yield point, an applied shear stress produces an elastic deformation that disappears when the applied stress is removed. However, when the applied shear stress is higher than the yield point, the structure of the fluid is destroyed, the sample deformation increases at a relatively rapid rate and, as a result, the sample flows [32]. Figure 2 shows the yield stress values of the tested doughs. The conducted statistical analysis showed a significant statistical impact of the type of nut oil cake, the level of replacing flour with oil cake and the interaction between these factors on the values of the described parameter. It was found that the yield stress value ranged from 1994 ± 74 Pa (WDWOC5%) to 4010 ± 101 Pa (WDHOC15%). The WDWOC5% dough did not differ significantly in this parameter from the control dough. However, all other doughs were characterized by higher flow limit values compared to the control dough. Moreover, in the case of both dough variants, an increase in the yield point value was observed with the increasing share (greater degree of flour substitution) of nut cakes. This may be due to the increasing content of ballast substances in such dough and the decreasing level of flour. In the case of doughs containing oil cakes, an increasing value of the yield point is observed. Therefore, these doughs became more and more stiff and needed more and more tangential stress to induce flow, a behavior typical of a fluid.

The WDHOC15% and WDWOC15% samples were also characterized by high values of the K′ and K″ parameters, as well as low values of the J_0_ and J_1_ parameters, which indicates a more durable and stiff structure of the dough. This is confirmed by the linear correlation values that were calculated for the yield stress values. There was a significant correlation between the yield stress (YS) parameter and extensographic indicators after 30 min, such as resistance, extensibility and ratio number (respectively: r = 0.85, r = −0.89, r = 0.97; *p* < 0.05), extensographic parameters after 60 min as extensibility and ratio number (r = −0.77, r = 0.90; *p* < 0.05) and extensographic parameters after 90 min as extensibility and ratio number (respectively: r = −0.83, r = 0.79; *p* < 0.05). A linear correlation was also shown with the parameters K′, K″, n″ and J_1_ (respectively: r = 0.84, r = 0.83, r = −0.81, r = −0.83; *p* < 0.05).

Knowledge about the rheological properties of dough is essential for optimizing the bread production process as it makes it possible to predict how the dough will behave during mixing, fermentation and forming loaves [33]. In the case of a compositionally complex system such as the one present in this work, non-starch substances are also involved in building the gel structure. Figure 3 shows the mechanical spectra representing the dependence of the storage modulus G′ and the loss modulus G″ on the angular velocity (ω) of sample doughs with the addition of hazelnut or walnut oil cakes. Table 4 contains the determined parameters of power-law models used to describe the determined experimental curves. Additionally, Figure 4 shows the relationship between the tangent of the phase shift angle (tan δ) as a function of the angular frequency (*ω*) determined for example systems. According to Dobraszczyk and Morgenstein [34], in the case of a cross-linked polymer network, the elastic properties (G′) generally prevail over the viscous properties (G″). The research showed that in all analyzed cases, the G′ module dominated over the G″ module. Therefore, the determined curves have a course characteristic of weak gels, with a clear dominance of elastic features over viscous features and additional values of the phase shift angle tangent (tan δ = G″/G′) greater than 0.1 [35]. This is consistent with the observations reported by Pycia and Juszczak [7]. In the wheat doughs containing ground hazelnuts or walnuts, the domination of the elastic properties over the viscous ones was confirmed. In the tested WDHOC and WDWOC doughs, a clear increase in the values of the G′ and G″ was observed with an increase in the share of oil cakes in the systems. Pycia and Juszczak [7] had opposite observations regarding the value of the G′ and G″ module in the case of the growing share of hazelnuts or walnuts in wheat doughs. According to Ungureanu-Iuga et al. [33] a small addition of flour from sprouted grains or lentils resulted in an increase in the G′ and G″ moduli in the wheat dough. Korus et al. [36] had similar observations in their study of gluten-free dough using resistant starch. The cited authors found that an increase in the content of resistant starch resulted in an increase in the G′ and G″ moduli, except for the mixture with the highest doses of RS, which had a small impact on the decrease in the G′ and G″ moduli. A similar tendency was observed by Lazaridou et al. [37] examining gluten-free dough with the addition of hydrocolloids and Sivaramakrishnan et al. [38] analyzing the influence of hydroxypropyl methylcellulose (HPMC) on the rheological properties of rice dough. However, Dokic et al. [31] found that the modular coefficient (tan. δ = G″/G′) decreased with the increase in the content of chestnut flour in the tested dough. Li et al. [18] analyzing the impact of enriching wheat dough with different gluten content with extruded bran, found an increase in the values of G′ and G″ moduli. This behavior of the system is probably due to the competition for water between the bran and the gluten network in the dough formation process, which made water uptake more difficult through the gluten network and caused intermolecular interaction between water and other dough ingredients in the reinforced dough system. Guardianelli et al. [26], analyzing the impact of the addition of pistachio flour (PBF) to wheat flour on the rheological properties of the dough, found that with the increase in the PBF content, it significantly increased dough stiffness and elasticity. The G′ and G″ moduli are reflected in the parameter values of the power equations used to describe these curves. The values of the constants K′ and K″ illustrate the values of the storage modulus (G′) and the loss modulus (G″), respectively, at an angular velocity of 1 rad/s [39].

The performed statistical analysis for the parameters K′ and K″ only confirmed a significant impact on their values of the level of the addition of nut oil cakes and the interaction between both tested factors. However, the type of oil cakes had no significant impact on the values of these parameters (*p* > 0.05). It was found that the value of the K′ constant reflecting the value of the G′ modulus (storage modulus) varied within a wide range, from 6677.7 ± 121.0 Pa∙s^n′^ (WDWOC5%) to 20,465.7 ± 694.6 Pa∙s^n′^ (WDWOC15%). It was shown that the average value of the K′ constant for WDHOC was 13,988.1 Pa∙s^n′^ and was higher than the control sample by 93%. However, in the case of WDWOC samples, the average value of the K′ parameter was 13,526.7 Pa∙s^n′^ and was 84% higher compared to the control sample. In both cases, an increase in the K′ value was observed as the share of oil cakes in the systems increased. The WDWOC5% sample was characterized by a lower value of the K′ constant compared to the control sample. A similar situation was observed in the case of the K″ constant reflecting the values of the loss modulus G″. It was shown that the average K″ value in the case of WDHOC was 5999.2 Pa∙s^n″^ and was higher than the control sample by 92%. Similarly, the average K″ value in the case of WDWOC was 58,885.6 and was 88% higher compared to the control sample. The wheat dough with the addition of hazelnuts or walnuts tested by Pycia and Juszczak [7] was characterized by lower values of K′ and K″ constants with similar shares of hazelnuts or walnuts. However, in this case, the authors observed a decrease in the values of the K′ and K″ parameters as the share of nuts in the system increased. Therefore, the differentiating factor is probably the fat and fiber present in opposite amounts in nuts and oil cakes. The performed two-way analysis confirmed that the tested factors generally did not have a significant impact on the value of the n′ and n″ constants. The analyzed samples generally did not differ statistically within these indicators. The values of these parameters ranged from 0.20 to 0.24. The obtained values of the parameters n′ and n″ are similar to those presented by Korus et al. [36] for gluten-free dough containing resistant starch and Adamczyk et al. [19] for rye–wheat dough supplemented with various fiber. Adamczyk et al. [19] also observed a statistically insignificant influence of various fiber preparations on the values of parameters n′ and n″ in the case of the tested rye–wheat dough and its mixtures. Therefore, as can be seen from the values of the parameters n′ and n″ of the rheological models describing the determined mechanical spectra, the differences in the dependences of the G′ modulus and G″ modulus on the angular velocity determined between the samples were very small. 

A significant correlation was found between K′ parameter and the dough development time (r = 0.77; *p* < 0.05) and extensographic parameters after 30 min such as resistance, maximum and ratio number (respectively: r = 0.94, r = 0.86, r = 0.78; *p* < 0.05); extensographic parameters after 60 min as resistance and maximum, (respectively: r = 0.82, r = 0.84; *p* < 0.05); and extensographic parameters after 90 min as resistance and maximum, (respectively: r= 0.78, r = 0.83; *p* < 0.05). A linear correlation was also shown between the values of the K″ parameter and the dough development time (r = 0.78; *p* < 0.05) and extensographic parameters after 30 min such as resistance, maximum and ratio number (r = 0.94, r = 0.85, r = 0.76, respectively; *p* < 0.05); extensographic parameters after 60 min as resistance and maximum, (respectively: r = 0.83, r = 0.85; *p* < 0.05); and extensographic parameters after 90 min as resistance and maximum, (respectively: r =0.77, r = 0.82; *p* < 0.05). The WDHOC15% and WDWOC15% samples were characterized by high values of the K′ and K″ parameters, as well as low values of the J_0_ and J_1_ parameters, which indicates a more durable and stiff structure of the dough. This is confirmed by the linear correlation values between these parameters and the yield point value. A linear correlation was also shown with the parameters K′, K″ and J_1_ (respectively: r = 0.84, r = 0.83, r = −0.83; *p* < 0.05).

### 3.4. Creep and Recovery Test

To comprehensively assess the viscoelastic properties of the dough, the creep and recovery test was also used. Selected experimental creep and recovery curves are shown in Figure 5. Table 5 contains the parameters of the Burgers model used to describe the experimental curves. As indicated by the high values of the R^2^ coefficient (approximately 0.9800), Burger’s model describes the determined experimental curves well. According to Steffe [39] and Gałkowska and Juszczak [40], when starting creep tests, the data obtained using Burgers’ mechanical analogue show the initial elastic reaction, which is illustrated by an immediate change in compliance under the influence of the spring in part of the Maxwell model. Then the Kelvin term, i.e., the combination of parallel springs, causes an exponential change in compliance related to the delay time. After a certain period of time, the independent distribution unit generates a purely viscous system response. The Maxwell and Kelvin-Voigt models represent simple mechanical viscoelastic models. In the case of this first model, the strain is the sum of the elastic strain and the viscous strain. This model is very useful for describing the relaxation phenomenon, but it is not suitable for describing the creep process, which is the second characteristic phenomenon for viscoelastic bodies. In turn, the Kelvin-Voigt model is not suitable for describing the phenomenon of stress relaxation, but it perfectly describes the phenomenon of creep, i.e., the increase in deformation over time of the body under the influence of load. The phenomenon of creep and recovery is the result of the reorientation of bonds in a viscoelastic material under the influence of load [41,42]. The Burgers model used to describe creep and recovery curves is a serial combination of Maxwell and Kelvin-Voigt elements [43]. The determined experimental curves have a course characteristic of a body with viscoelastic properties. It was found that the control dough was characterized by the greatest susceptibility to deformation. However, the presence of both hazelnut oil cakes (Figure 5a) and walnut oil cakes (Figure 5b) reduced the dough’s susceptibility to deformation the greater their share. A similar tendency was observed by Adamczyk et al. [19]. The cited authors noted that the addition of various preparations rich in fiber to the mixture of rye–wheat flour significantly reduced susceptibility to deformation. Korus et al. [36] made similar observations in their research on gluten-free dough. The cited authors stated that the replacement of starch with resistant starch (RS) preparations and mixes diminished their deformation compliance. The lowest deformation compliance was exhibited by formulations with addition of tapioca RS starch. The opposite behavior of dough containing hazelnuts or walnuts was observed in previous work [7]. It was noted that the presence of nuts increased the susceptibility of wheat dough to deformation. This dependence deepened as their share in the dough increased. As part of the analysis, the immediate compliance (J_0_) was determined, which reflects the energy of the elastic stretching of bonds after applying stress. This energy disappears immediately after the stress is removed. In turn, viscoelastic compliance (J_1_) is related to the cleavage and conversion of chemical bonds [40,41,42].

The analysis of variance showed that the value of immediate susceptibility depended statistically significantly only on the level of the share of nut oil cakes in the dough. The type of oil cakes and interactions between the tested factors were insignificant for this parameter. It was shown that the highest value of immediate susceptibility J_0_ was characteristic of wheat dough without the addition of oil cakes. However, their addition reduced the value of this parameter. Because the average value of J_0_ in the case of WDHOC and WDWOC was on average 0.8 × 10^−4^ Pa^−1^. Pycia and Juszczak [7] had opposite observations. The cited authors noticed that wheat doughs containing ground hazelnuts and walnuts were characterized by higher J_0_ values compared to the control sample. It was similar in the case of the J_1_ parameter. However, in the described research, the control sample had the highest value (J_1_), and doughs containing nut oil cakes had a lower value of this parameter. In the case of viscoelastic compliance (J_1_), all tested factors significantly influenced the value of this parameter (*p* < 0.05). Therefore, the average value of this parameter for WDHOC was on average 1.8 × 10^−4^ Pa^−1^, and for WDWOC 2.0 × 10^−4^ Pa^−1^. In both variants, the value of viscoelastic compliance decreased significantly as the share of nut oil cakes increased (*p* < 0.05). Dokic et al. [31] found that the dough compliance (J) determined by creep and recovery curves decreased, thus more chestnut flour resulted in a more brittle consistency of the dough. The presence of nut oil cakes had a significant effect on the zero shear viscosity (η_0_). It was shown that this parameter decreased significantly as the level of OC in the wheat dough increased. Therefore, the highest value of this parameter was observed in doughs with a 15% content of oil cakes (Table 5). The viscoelastic compliance (J_1_) is related to the rupture and conversion of the bond. The delay time (λ_ret_) describes the delayed elastic response of a viscoelastic material to an applied stress [40,42]. It was found that the retardation time values for WDHOC decreased with the increase in the degree of substitution of flour with nut oil cake, while for systems with WDWOC the relationship was reverse. WDWOC15% dough was characterized by the highest retardation time value. Atudorei et al. [44] analyzed the possibility of using sprouted and dried soybeans as an addition to the main ingredients intended for bread production. The cited authors demonstrated a significant effect of the addition of 5%, 10%, 15% and 20% sprouted soy flour (GSF) on dough rheology and bread quality. The GSF addition had the effect of decreasing the values of the creep and recovery compliances. At the same time, the retardation time was increased. 

A linear correlation was found between the values of the J_0_ parameter and extensographic parameters after 30 min such as resistance, maximum and ratio number (respectively: r = −0.90, r = −0.86, r = −0.79; *p* < 0.05); extensographic parameters after 60 min as maximum (r = −0.77; *p* < 0.05); and extensographic indicators after 90 min as resistance and ratio number (r = −0.77, r = −0.78; *p* < 0.05,). A linear correlation was also demonstrated with the parameters K′, K″ and n″ (r = −0.84, r = −0.84, r = 0.83; *p* < 0.05). However, the J_1_ parameter correlated linearly with extensographic parameters after 30 min such as resistance, maximum and ratio number (respectively: r = −0.99, r = −0.93, r = −0.84; *p* < 0.05); extensographic parameters after 60 min as resistance, maximum and ratio number (r = −0.84, r = −0.86, r = −0.76; *p* < 0.05); and extensographic parameters after 90 min such as resistance, maximum and ratio number (respectively: r = −0.87, r= −0.76, r = −0.92; *p* < 0.05). There was also a linear correlation of the described parameter with the parameters K′, K″, n″, J_0_, η_0_ and the yield stress (respectively: r = −0.93, r = −0.92, r = 0.90, r = −0.83; *p*< 0.05). A linear correlation was also shown between the eta0 parameter and the dough development time (r = 0.85; *p* < 0.05) and extensographic parameters after 30 min, such as resistance and maximum (respectively: r = 0.91, r = 0.85, *p* < 0.05); extensographic parameters after 60 min as resistance and maximum (r = 0.86, r = 0.92; *p* < 0.05); extensographic parameters after 90 min as resistance and ratio number (respectively: r = 0.78, r = 0.78; *p* < 0.05); and parameters K′, K″ and n″ (respectively: r = 0.97, r = 0.98, r = −0.91; *p* < 0.05).

## 4. Conclusions

The results indicate that the mechanical properties of the dough depend significantly on the selection of the type and amount of recipe ingredients. The application possibilities of hazelnut or walnut oil cakes examined in this work are an interesting direction in the use of these by-products of the food industry. It was found that the presence of oil cakes in wheat dough significantly affects its rheological properties (farinographic, extensographic and viscoelastic). The dough containing oil cakes had a more durable and stiffer consistency and was less susceptible to deformation. Therefore, it differed significantly from wheat dough without additives. However, the stiff and deformation-resistant structure of the dough increased with the increase in the share of nut oil cakes. However, it should be remembered that the rheological properties of the dough and its consistency determine the quality of the product after baking. Therefore, the influence of the share of nut oil cakes on baking parameters and the quality of the obtained bread requires further research. Hazelnut or walnut oil cakes are a valuable raw material in terms of chemical composition because they are a rich source of fiber, protein, minerals and a number of bioactive substances, so they can also improve or shape the health-promoting properties of bread, and can also shape the texture of the bread crumb during storage. Nevertheless, this research topic requires continuation. Moreover, this type of food use of oil cake flour, apart from enriching food, in this case bakery products, fits perfectly into the zero waste trend, i.e., waste-free production, and is also suitable for vegans and can provide products rich in vegetable protein on a good scale and amino acids.

## Figures and Tables

**Figure 1 foods-13-00140-f001:**
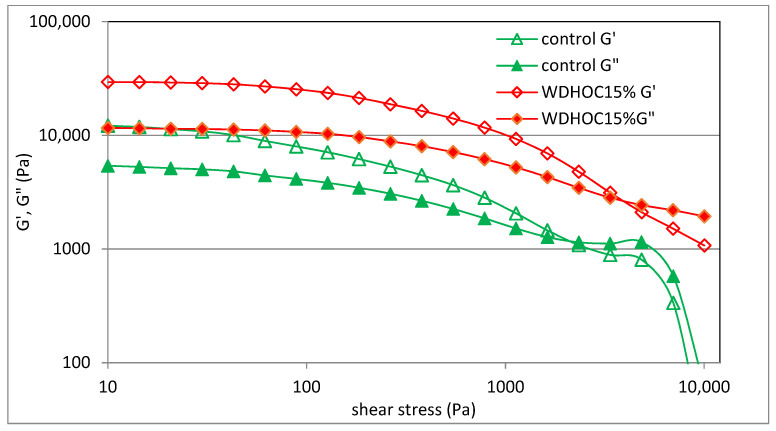
Dependence of the G′ and G″ moduli on the shear stress of selected samples (WDHOC—dough with hazelnut oil cakes).

**Figure 2 foods-13-00140-f002:**
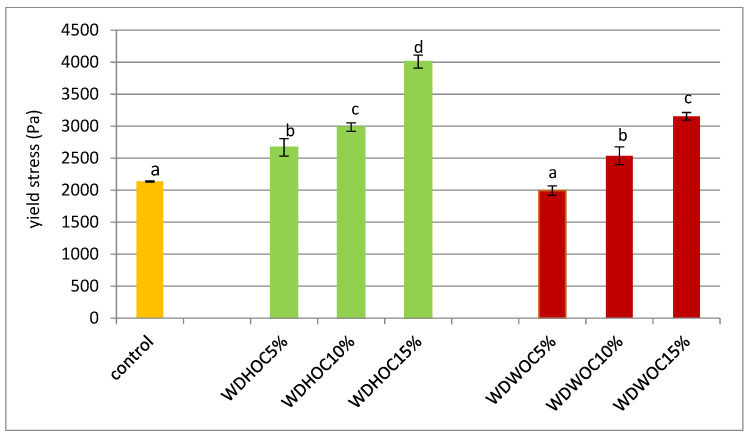
Yield stress of doughs based on wheat flour with the use of nut oil cakes. WDHOC/WDWOC—dough with hazelnut/walnut oil cakes. The average values ± SD in individual columns marked with the same superscripts do not differ statistically significantly (*p* < 0.05).

**Figure 3 foods-13-00140-f003:**
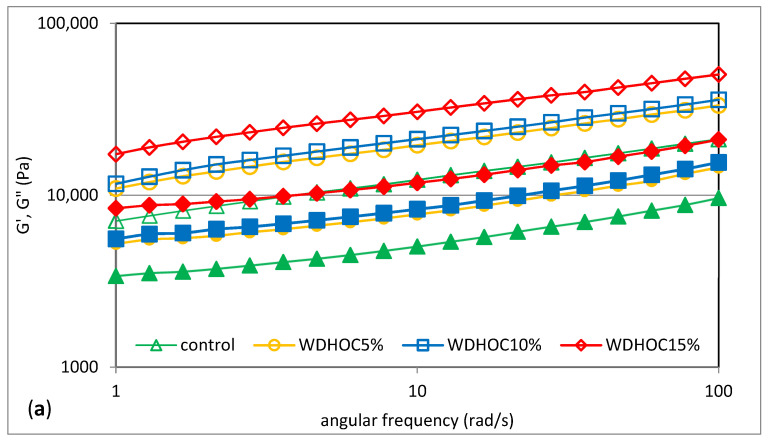
Frequency sweep curves of doughs with hazelnuts oil cake (**a**) or walnuts oil cake (**b**). G′—empty marker; G″—filled markers. WDHOC/WDWOC—dough with hazelnut/walnut oil cakes.

**Figure 4 foods-13-00140-f004:**
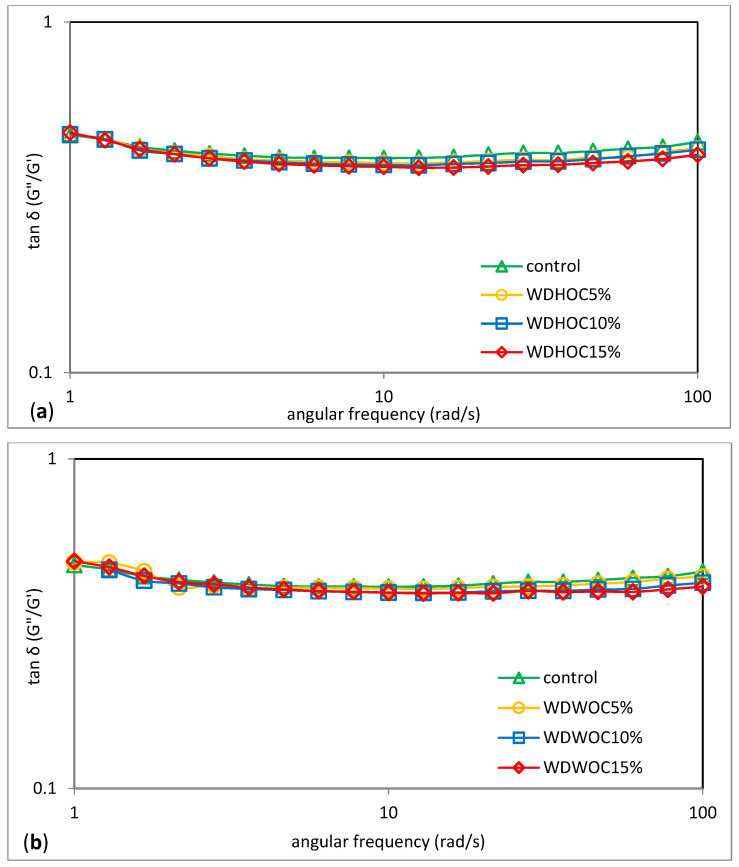
Tangent δ depending on the angular frequency of the doughs with hazelnuts (**a**) or walnuts (**b**) oil cakes. WDHOC/WDWOC—dough with hazelnut/walnut oil cakes.

**Figure 5 foods-13-00140-f005:**
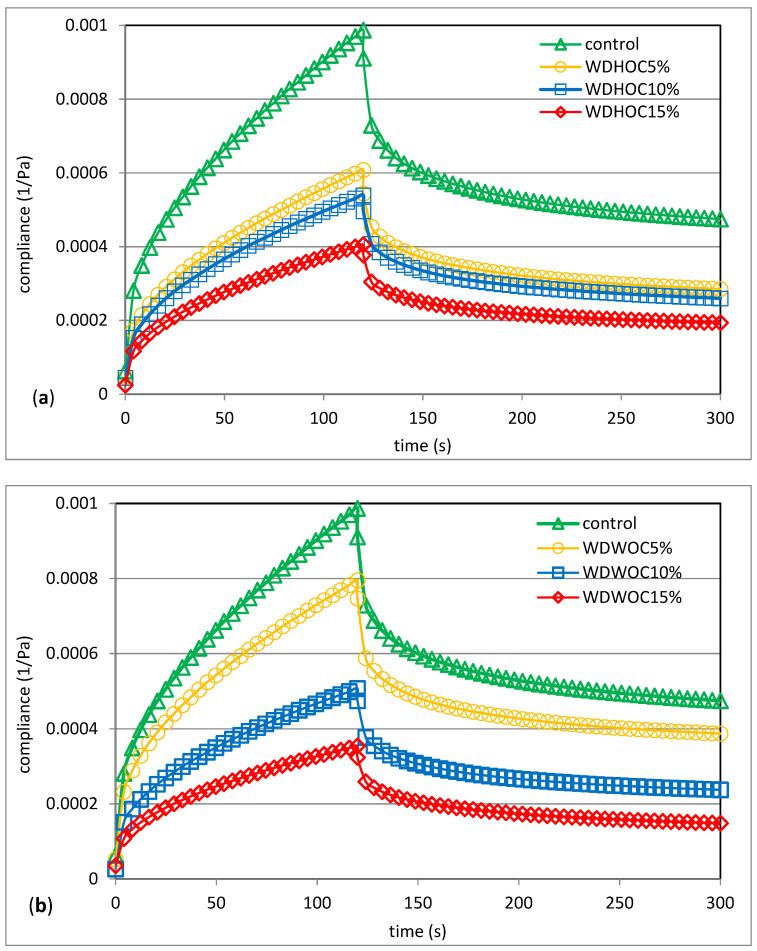
Creep and recovery curves of control sample and dough with hazelnuts (**a**) or walnuts (**b**) oil cakes. WDHOC/WDWOC—dough with hazelnut/walnut oil cakes.

**Table 1 foods-13-00140-t001:** Farinographic parameters of the examined samples with various proportions of nut oil cakes.

Sample	Water Absorption [%]	Dough Development Time [min]	Dough Stability [min]	Degree of Softening [BU]	Farinograph Number
control	52.3 ± 1.0 ^a^	3.5 ± 1.1 ^a^	6.1 ± 0.6 ^a^	64.0 ± 16.5 ^d^	67.7 ± 7.8 ^a^
WDHOC5%	52.7 ± 0.3 ^a^	5.1 ± 0.1 ^bc^	7.5 ± 0.2 ^b^	35.7 ± 2.1 ^b^	89.0 ± 3.6 ^c^
WDHOC10%	53.8 ± 0.9 ^b^	4.3 ± 0.5 ^b^	5.9 ± 0.5 ^a^	67.0 ± 11.1 ^d^	74.3 ± 8.4 ^b^
WDHOC15%	52.8 ± 0.3 ^a^	4.7 ± 0.4 ^bc^	5.5 ± 0.4 ^a^	54.0 ± 4.6 ^c^	73.0 ± 3.0 ^b^
WDWOC5%	53.6 ± 0.2 ^ab^	4.2 ± 1.2 ^b^	9.0 ± 0.8 ^c^	31.7 ± 7.6 ^b^	94.0 ± 14.0 ^c^
WDWOC10%	53.7 ± 0.4 ^b^	6.0 ± 0.0 ^c^	13.6 ± 0.2 ^d^	17.3 ± 7.2 ^a^	122.3 ± 7.8 ^d^
WDWOC15%	53.8 ± 0.3 ^b^	7.5 ± 1.3 ^d^	13.7 ± 0.3 ^d^	7.3 ± 2.9 ^a^	150.0 ± 0.0 ^e^
	two-way ANOVA *p*-values				
Factor 1	*p* < 0.001	*p* < 0.001	*p* < 0.001	*p* < 0.001	*p* < 0.001
Factor 2	*p* = 0.086	*p* < 0.001	*p* < 0.001	*p* < 0.001	*p* < 0.001
Factor 1 × factor 2	*p* = 0.136	*p* < 0.001	*p* < 0.001	*p* < 0.001	*p* < 0.001

WDHOC/WDWOC—dough with hazelnut/walnut oil cakes; factor 1—type of oil cake; factor 2—substitution level. The average values ± SD in individual columns marked with the same superscripts do not differ statistically significantly (*p* < 0.05).

**Table 2 foods-13-00140-t002:** Extensographic properties of the tested wheat doughs containing nut oil cakes.

Sample	Energy [cm^2^]			Resistance to Extension [BU]			Extensibillity [mm]		
	30 min	60 min	90 min	30 min	60 min	90 min	30 min	60 min	90 min
control	88.0 ± 5.7 ^bc^	82.5 ± 3.5 ^ab^	80.5 ± 4.9 ^ab^	324.0 ± 1.4 ^a^	341.5 ± 19.1 ^a^	334.5 ± 14.8 ^a^	159.0 ± 14.1 ^e^	145.5 ± 2.1 ^d^	147.0 ± 0.0 ^d^
WDHOC5%	98.0 ± 1.4 ^c^	98.5 ± 3.5 ^bc^	97.5 ± 2.1 ^b^	463.0 ± 9.9 ^b^	538.0 ± 7.1 ^c^	604.0 ± 21.2 ^bcd^	133.5 ± 0.7 ^c^	122.5 ± 3.5 ^c^	111.0 ± 1.4 ^b^
WDHOC10%	82.0 ± 2.8 ^b^	87.0 ± 5.7 ^ab^	83.5 ± 2.1 ^ab^	477.0 ± 15.6 ^b^	537.0 ± 32.5 ^c^	577.5 ± 30.4 ^bc^	115.0 ± 0.0 ^b^	111.5 ± 0.7 ^b^	104.0 ± 1.4 ^ab^
WDHOC15%	70.0 ± 2.8 ^a^	74.5 ± 3.5 ^a^	68.0 ± 5.7 ^a^	532.0 ± 5.7 ^c^	578.0 ± 9.9 ^c^	553.0 ± 31.1 ^b^	94.0 ± 1.4 ^a^	91.5 ± 2.1 ^a^	93.5 ± 12.0 ^a^
WDWOC5%	89.0 ± 0.0 ^bc^	103.5 ± 3.5 ^c^	125.5 ± 4.9 ^c^	346.0 ± 5.7 ^a^	426.5 ± 20.5 ^b^	311.5 ± 2.1 ^a^	152.0 ± 4.2 ^e^	147.5 ± 6.4 ^d^	191.0 ± 1.4 ^e^
WDWOC10%	110.0 ± 5.7 ^d^	131.5 ± 9.2 ^d^	133.0 ± 15.6 ^c^	468.0 ± 17.0 ^b^	401.0 ± 67.9 ^b^	669.0 ± 59.4 ^cd^	141.5 ± 2.1 ^d^	174.0 ± 7.1 ^e^	131.0 ± 1.4 ^c^
WDWOC15%	120.5 ± 7.8 ^d^	133.5 ± 6.4 ^d^	122.0 ± 4.2 ^c^	541.0 ± 19.8 ^c^	673.0 ± 25.5 ^d^	675.5 ± 46.0 ^d^	141.5 ± 3.5 ^d^	129.0 ± 2.8 ^c^	117.0 ± 4.2 ^b^
	two-way ANOVA *p*-values								
Factor 1	*p* < 0.001	*p* < 0.001	*p* < 0.001	*p* < 0.001	*p* < 0.001	*p* = 0.260	*p* < 0.001	*p* < 0.001	*p* < 0.001
Factor 2	*p* = 0.713	*p* = 0.197	*p* = 0.041	*p* < 0.001	*p* < 0.001	*p* < 0.001	*p* < 0.001	*p* < 0.001	*p* < 0.001
Factor 1 × factor 2	*p* < 0.001	*p* < 0.001	*p* = 0.095	*p* < 0.001	*p* < 0.001	*p* < 0.001	*p* < 0.001	*p* < 0.001	*p* < 0.001

WDHOC/WDWOC—dough with hazelnut/walnut oil cakes; factor 1—type of oil cake; factor 2—substitution level. The average values ± SD in individual columns marked with the same superscripts do not differ statistically significantly (*p* < 0.05).

**Table 3 foods-13-00140-t003:** Extensographic properties of the tested wheat doughs containing nut oil cakes.

Sample	Maximum [BU]			Ratio Number		
	30 min	60 min	90 min	30 min	60 min	90 min
control	398.0 ± 5.7 ^a^	405.5 ± 21.9 ^a^	400.0 ± 14.1 ^a^	2.1 ± 0.2 ^a^	2.4 ± 0.2 ^a^	2.3 ± 0.1 ^b^
WDHOC5%	530.0 ± 4.2 ^cd^	593.0 ± 18.4 ^c^	646.0 ± 9.9 ^c^	3.5 ± 0.1 ^bc^	4.5 ± 0.1 ^b^	5.5 ± 0.2 ^c^
WDHOC10%	494.5 ± 19.1 ^c^	552.5 ± 37.5 ^bc^	586.0 ± 26.9 ^bc^	4.2 ± 0.1 ^d^	4.8 ± 0.3 ^bc^	5.6 ± 0.4 ^c^
WDHOC15%	532.5 ± 6.4 ^cd^	579.5 ± 9.2 ^bc^	553.5 ± 31.8 ^bc^	5.7 ± 0.1 ^e^	6.3 ± 0.0 ^d^	5.9 ± 0.4 ^c^
WDWOC5%	417.5 ± 9.2 ^b^	503.5 ± 7.8 ^b^	485.5 ± 16.3 ^b^	2.3 ± 0.1 ^a^	2.9 ± 0.3 ^a^	1.6 ± 0.0 ^a^
WDWOC10%	542.0 ± 17.0 ^cd^	548.5 ± 58.7 ^bc^	770.0 ± 67.9 ^d^	3.4 ± 0.1 ^b^	2.4 ± 0.5 ^a^	5.2 ± 0.5 ^c^
WDWOC15%	561.5 ± 46.0 ^d^	737.0 ± 17.0 ^d^	764.0 ± 62.2 ^d^	3.8 ± 0.3 ^c^	5.2 ± 0.3 ^c^	5.8 ± 0.6 ^c^
	two-way ANOVA *p*-values					
Factor 1	*p* = 0.381	*p* = 0.272	*p* < 0.001	*p* < 0.001	*p* < 0.001	*p* < 0.001
Factor 2	*p* < 0.001	*p* < 0.001	*p* < 0.001	*p* < 0.001	*p* < 0.001	*p* < 0.001
Factor 1 × factor 2	*p* < 0.001	*p* < 0.001	*p* < 0.001	*p* < 0.001	*p* = 0.398	*p* < 0.001

WDHOC/WDWOC—dough with hazelnut/walnut oil cakes; factor 1—type of oil cake; factor 2—substitution level. The average values ± SD in individual columns marked with the same superscripts do not differ statistically significantly (*p* < 0.05).

**Table 4 foods-13-00140-t004:** Parameters of power law models designated for frequency sweep curves of examined samples.

Sample	K′ [Pa s^n′^]	n′	R^2^	K″ [Pa s^n″^]	n″	R^2^
control	7347.3 ± 515.6 ^b^	0.23 ± 0.01 ^ab^	0.9987	3121.0 ± 75.6 ^b^	0.23 ± 0.01 ^b^	0.9827
WDHOC5%	11,382.3 ± 479.0 ^c^	0.23 ± 0.01 ^ab^	0.9980	4898.7 ± 37.2 ^c^	0.22 ± 0.01 ^b^	0.9820
WDHOC10%	12,347.3 ± 717.4 ^c^	0.23 ± 0.00 ^ab^	0.9957	5268.7 ± 278.6 ^c^	0.22 ± 0.01 ^b^	0.9840
WDHOC15%	18,234.7 ± 525.2 ^e^	0.22 ± 0.01 ^a^	0.9967	7830.3 ± 290.0 ^e^	0.20 ± 0.00 ^a^	0.9783
WDWOC5%	6677.7 ± 121.0 ^a^	0.24 ± 0.01 ^b^	0.9957	2907.3 ± 61.5 ^a^	0.23 ± 0.01 ^b^	0.9730
WDWOC10%	13,436.7 ± 819.0 ^d^	0.22 ± 0.01 ^a^	0.9977	5749.3 ± 347.5 ^d^	0.20 ± 0.00 ^a^	0.9847
WDWOC15%	20,465.7 ± 694.6 ^f^	0.23 ± 0.01 ^ab^	0.9963	9000.3 ± 313.3 ^f^	0.20 ± 0.01 ^a^	0.9900
	two-way ANOVA *p*-values					
Factor 1	*p* = 0.130	*p* = 0.483		*p* = 0.360	*p* = 0.839	
Factor 2	*p* < 0.001	*p* = 0.022		*p* < 0.001	*p* < 0.001	
Factor 1 × factor 2	*p* < 0.001	*p* = 0.079		*p* < 0.001	*p* = 0.042	

WDHOC/WDWOC—dough with hazelnut/walnut oil cakes; factor 1—type of oil cake; factor 2—substitution level. The average values ± SD in individual columns marked with the same superscripts do not differ statistically significantly (*p* < 0.05).

**Table 5 foods-13-00140-t005:** Parameters of the Burgers model for creep and recovery curves.

Sample	J_0_ (×10^−4^) (Pa^−1^)	J_1_ (×10^−4^) (Pa^−1^)	η_0_ (×10^4^) (Pa·s)	λ_ret_ (s)	R^2^
control	1.5 ± 0.2 ^e^	3.2 ± 0.2 ^e^	24.1 ± 1.0 ^a^	18.3 ± 1.6 ^ab^	0.9813
WDHOC5%	1.1 ± 0.0 ^d^	2.1 ± 0.1 ^c^	40.8 ± 1.7 ^b^	24.2 ± 2.0 ^cd^	0.9829
WDHOC10%	0.9 ± 0.0 ^c^	1.8 ± 0.0 ^b^	43.4 ± 3.3 ^b^	22.3 ± 3.0 ^bcd^	0.9818
WDHOC15%	0.6 ± 0.0 ^a^	1.4 ± 0.0 ^a^	58.0 ± 1.8 ^c^	20.8 ± 1.9 ^abc^	0.9797
WDWOC5%	1.2 ± 0.1 ^d^	2.8 ± 0.0 ^d^	29.1 ± 2.0 ^a^	16.8 ± 1.8 ^a^	0.9802
WDWOC10%	0.7 ± 0.0 ^b^	1.8 ± 0.0 ^b^	47.2 ± 2.9 ^b^	18.9 ± 2.3 ^ab^	0.9813
WDWOC15%	0.6 ± 0.0 ^a^	1.3 ± 0.0 ^a^	75.5 ± 7.5 ^d^	26.2 ± 2.6 ^d^	0.9818
	two-way ANOVA *p*-values				
Factor 1	*p* = 0.623	*p* < 0.001	*p* = 0.103	*p* = 0.120	
Factor 2	*p* < 0.001	*p* < 0.001	*p* < 0.001	*p* = 0.070	
Factor 1 × factor 2	*p* = 0.056	*p* < 0.001	*p* < 0.001	*p* < 0.001	

WDHOC/WDWOC—dough with hazelnut/walnut oil cakes; factor 1—type of oil cake; factor 2—substitution level. The average values ± SD in individual columns marked with the same superscripts do not differ statistically significantly (*p* < 0.05).

## Data Availability

Data is contained within the article.

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
