# Peer review of "Rheological Characteristics of Wheat Dough Containing Powdered Hazelnuts or Walnuts Oil Cakes"

_foods, 2023, doi:10.3390/foods13010140_

Round 1
Reviewer 1 Report
Comments and Suggestions for Authors
This paper determined the rheological properties of the wheat dough containing powdered hazelnuts or walnuts oil cakes using a brabender farinograph, a brabender extensograph and an osicillaotory rheometer. The results were shown clearly and detailedly. Several questions should be considered,
1) The biochemical composition of powdered hazelnuts or walnuts oil cakes should be given. In the conclusion section and abstract section, the conclusion was not clearly and specifically given.
2) What is the aim of this study? How to use the results of this study in the practice?
3) The letters in tables should be noted.
4) What is the difference between theological characteristics of wheat dough containing powdered hazelnuts and walnuts oil cakes? Which one is better for use?
Author Response
Reviewer 1
This paper determined the rheological properties of the wheat dough containing powdered hazelnuts or walnuts oil cakes using a brabender farinograph, a brabender extensograph and an osicillaotory rheometer. The results were shown clearly and detailedly. Several questions should be considered,
The biochemical composition of powdered hazelnuts or walnuts oil cakes should be given.
Response: Information on the basic composition of wheat flour and hazelnut and walnut oil cakes has been supplemented. Therefore, appropriate information has been added to the research material section.
In the conclusion section and abstract section, the conclusion was not clearly and specifically given.
Response: The abstract and conclusions of the work describe the effect of the addition of nut oil cakes on the rheological properties of wheat dough. These sections were improved.
What is the aim of this study? How to use the results of this study in the practice?
Response: The aim of this work was indicated in the abstract and introductions. Moreover, the introduction to the work describes the reasons why knowledge about the rheological properties of dough is important. This is important because it affects the behavior of the dough during fermentation, dividing or forming loaves, but also for assessing the quality of the finished product after baking. Therefore, the presented research results have a practical dimension, as described in the manuscript. The results obtained during these tests show the impact of the addition of peanut cake flour as an ingredient enriching bakery products on the rheological and mechanical characteristics of the dough.
The letters in tables should be noted.
Response: All tables have been checked. Superscript letters placed next to the mean values of individual parameters symbolize the occurrence of statistically significant differences at the appropriate level of significance.
What is the difference between theological characteristics of wheat dough containing powdered hazelnuts and walnuts oil cakes? Which one is better for use?
Response: The aim of the study was to assess the influence of the addition of walnut and hazelnut cakes on the rheological properties of wheat dough. The obtained test results prove the significant impact of these ingredients. The results and discussion section describes in detail the direction of these changes and attempts to explain the causes of these changes. It is difficult to determine whether the direction of these changes is always beneficial. Because all the tested rheological features changed. The dough containing oil cakes had a longer development time, a lower degree of softening, and was more stiff and resistant to deformation. In our opinion, it is impossible to clearly answer the question of which oil cake is better. Because of the similar composition chemical changes in the content of basic nutrients will be similar, and differences related to the origin of the cake may concern, for example, the composition of fatty acids, the content of minerals or vitamins, which should not have a significant impact on the physical characteristics of the dough. However, the observed differences in the rheological characteristics of dough may result from different contents, e.g. of individual protein or fiber fractions, which may modify the ability to absorb and bind water and thus influence the rheological characteristics.

Reviewer 2 Report
Comments and Suggestions for Authors
This study was focused on the addition of HOC and WOC instead of the WF. It gives us a new insight into the utilization of oil cakes. This study is mainly used to illustrate the physicochemical properties of the alternative dough, resulting in a lack of innovation.
However, there are still some basic compositions in HOC and WOC that the author does not clear. It should give the composition of two cakes. Because the starch, protein, oil, and dietary fiber content should influence the rheological properties. The main conclusions of the authors' tests are closely related to the specific composition of the dough. It is necessary to be added to the materials. Secondly, in Line 110, it may be the Power Law model.
Comments on the Quality of English Language
good
Author Response
Reviewer 2
This study was focused on the addition of HOC and WOC instead of the WF. It gives us a new insight into the utilization of oil cakes. This study is mainly used to illustrate the physicochemical properties of the alternative dough, resulting in a lack of innovation.
Response: Oil cakes based on walnuts and hazelnuts are still relatively rarely used in baking or confectionery practice. They are a valuable by-product in the food industry, rich in protein, fiber and minerals. Therefore, it is worth looking for their applications in the food industry. The recipe composition of the dough determines its rheological properties. Therefore, the situation is similar in the case of the addition of nut cakes, i.e. replacing part of the flour in the recipe with this factor. In turn, the rheological properties of the dough are important from the point of view of the fermentation process and the formation of dough pieces, as well as the quality of the finished product. There are no such studies in the literature describing the impact of nut cakes. So we hope that this will expand our knowledge in this area.
However, there are still some basic compositions in HOC and WOC that the author does not clear. It should give the composition of two cakes. Because the starch, protein, oil, and dietary fiber content should influence the rheological properties. The main conclusions of the authors' tests are closely related to the specific composition of the dough. It is necessary to be added to the materials. Secondly, in Line 110, it may be the Power Law model.
Response: The sentence in line 110 was corrected. Information on the basic composition of wheat flour and hazelnut and walnut oil cakes has been supplemented. Therefore, appropriate information has been added to the research material section.

Reviewer 3 Report
Comments and Suggestions for Authors
The use of agricultural byproducts, hazelnut and walnut cakes, in the replacement of wheat is valuable even though it has been studied for decades. A significant amount of data was gathered for the study. However, there are major problems with the presentation and terminology used in the manuscript. I suggest re-writing the manuscript and improve the figures and the language. Please consider the following points.
1. The abstract is too long; it should be around 300 words at most. You may change the first sentence as follows to improve clarity.
This study assessed edible oil industry byproducts, hazelnut and walnut cakes, to replace wheat flour dough based on farinograph and extensograph parameters and rheological measurements.
2. The abstract should state the statistical significance of the results. For example, lines 17-19. If the difference is significant, (P<0.05) needs to be added at the end of the sentence; otherwise, put (P>0.05).
3. Line 13. Replace "rheological properties" with "mechanical properties".
4. Avoid unnecessary sentences such as "the control sample was native wheat dough". This can be explained when introducing the samples in the methodology.
5. The proximate composition should be given for the ingredients used in the materials and methods section.
6. Yield stress is also a mechanical property; why do the authors exclude that from section 2.2.3. It should be merged with the rheological properties.
7. It is unnecessary to mention the methodology in the results. For example, Lines 142-146. Consider removing methodology information from the results.
Comments on the Quality of English Language
The language should be improved for clarity.
Author Response
Reviewer 3
The use of agricultural byproducts, hazelnut and walnut cakes, in the replacement of wheat is valuable even though it has been studied for decades. A significant amount of data was gathered for the study. However, there are major problems with the presentation and terminology used in the manuscript. I suggest re-writing the manuscript and improve the figures and the language.
Response: Thank you for your opinion. The manuscript has been revised according to the suggestions of many reviewers and the editor. We hope it is improved in its current form.
Please consider the following points.
- The abstract is too long; it should be around 300 words at most. You may change the first sentence as follows to improve clarity.
This study assessed edible oil industry byproducts, hazelnut and walnut cakes, to replace wheat flour dough based on farinograph and extensograph parameters and rheological measurements.
Response: The abstract has been corrected and shortened according to the reviewers' comments.
- The abstract should state the statistical significance of the results. For example, lines 17-19. If the difference is significant, (P<0.05) needs to be added at the end of the sentence; otherwise, put (P>0.05).
Response: The additional informations were added to the abstract section.
- Line 13. Replace "rheological properties" with "mechanical properties".
Response: As a result of improving the abstract, this phrase was removed. The abstract has been rewritten and improved.
- Avoid unnecessary sentences such as "the control sample was native wheat dough". This can be explained when introducing the samples in the methodology.
Response: Thank you for this attention. This sentence has been removed from the abstract section. However, it is placed in the research material section.
- The proximate composition should be given for the ingredients used in the materials and methods section.
Response: Information on the basic composition of wheat flour and hazelnut and walnut oil cakes has been supplemented. Therefore, appropriate information has been added to the research material section.
- Yield stress is also a mechanical property; why do the authors exclude that from section 2.2.3. It should be merged with the rheological properties.
Response: The separation of these two analyzes was dictated by the desire to arrange the results in a more transparent way. However, I agree with the reviewer's opinion that yield stress belongs to mechanical properties. Therefore, these two sections were combined. A similar approach was taken in the Results and Discussion section.
- It is unnecessary to mention the methodology in the results. For example, Lines 142-146. Consider removing methodology information from the results.
Response: Unnecessary information about the methodology has been removed from the results and discussion section.

Reviewer 4 Report
Comments and Suggestions for Authors
This manuscript describes a complex study with great potential to advance knowledge in the field. This makes an important contribution to the circular economy concept, as edible oil cake is a valuable by-product, rich in protein, fibre, minerals, vitamins and other bioactive substances, with great potential to improve the nutritional value of bakery products.
The paper is scientifically sound, based on in-depth and well-conducted research.
The abstract is well written and reflects the content of the paper. The introduction provides a solid background to the topic. The objective of the study is clearly stated.
The methodology needs to be improved with details on the preliminary processing of raw hazelnut oil cake and walnut oil cake obtained in the cold pressing process to obtain the powder of these by-products. It is very important to mention whether these raw by-products were used immediately after production or dried beforehand? Also, please provide the recipe and how the dough was obtained. Please include this data in section 2.1. Research material.
The results are clearly presented and well discussed. In tables and figures it is advisable to describe the abbreviations used for samples.
The conclusions contain some important key findings derived from the results. The data presented in Figures 1-4 and Tables 1-4 fully support every detail discussed in the paper.
The references cited are relevant to this research topic. Please note the journal's requirements for writing references.
Author Response
Reviewer 4
This manuscript describes a complex study with great potential to advance knowledge in the field. This makes an important contribution to the circular economy concept, as edible oil cake is a valuable by-product, rich in protein, fibre, minerals, vitamins and other bioactive substances, with great potential to improve the nutritional value of bakery products.
The paper is scientifically sound, based on in-depth and well-conducted research.
Response: Thank you very much for this review.
The abstract is well written and reflects the content of the paper. The introduction provides a solid background to the topic. The objective of the study is clearly stated.
Response: Thank you very much for this review.
The methodology needs to be improved with details on the preliminary processing of raw hazelnut oil cake and walnut oil cake obtained in the cold pressing process to obtain the powder of these by-products. It is very important to mention whether these raw by-products were used immediately after production or dried beforehand? Also, please provide the recipe and how the dough was obtained. Please include this data in section 2.1. Research material.
Response:The research presented in this manuscript constitutes a research topic, some of the results of which have already been published in the journal Foods. Nut oil cakes were described in detail in an earlier manuscript. This work was cited in the research material chapter. Therefore, in our opinion, there is no need to repeat this information. The dough for farinographic and extensographic examinations was made in a farinograph. For rheological analysis the dough samples were obtained by thoroughly mixing the wheat flour or its mix-tures with nut oil cakes with the amount of water determined by farinographic analysis. Fresh dough was prepared for each analysis. The additional informations were completed in manuscript.
The results are clearly presented and well discussed. In tables and figures it is advisable to describe the abbreviations used for samples.
Response: the additional informations about used abbreviations were added to the tables and figures
The conclusions contain some important key findings derived from the results. The data presented in Figures 1-4 and Tables 1-4 fully support every detail discussed in the paper.
Response: Thank you very much for this review.
The references cited are relevant to this research topic. Please note the journal's requirements for writing references.
Response: Thank you very much for this review. The references were checked.

Reviewer 5 Report
Comments and Suggestions for Authors
Title: The title effectively conveys the focus of the study: analyzing the rheological properties of wheat dough with hazelnut or walnut oil cakes. It accurately reflects the main components and scope of the research.
Abstract: The abstract provides a concise yet informative overview of the study. However, it could benefit from a more explicit articulation of the hypothesis or specific research question guiding the study.
Introduction: The introduction offers a comprehensive background on dough rheology and the potential impact of adding hazelnut or walnut oil cakes. The introduction offers contextual information but lacks a clearly defined research problem or a well-stated hypothesis. A stronger articulation of the research problem, along with a well-defined hypothesis, would greatly enhance the manuscript's foundation.
Materials and Methods: This section provides detailed information about the materials and methodologies employed. However, it lacks explicit connection to the study's aims or hypotheses. Consider organizing the section into subsections based on the experimental procedures. Create clear headings for different experimental techniques or analyses conducted (e.g., Farinograph Analysis, Extensograph Analysis, Oscillatory Rheometer Testing).
Results: The results section presents extensive data and analyses related to the rheological properties of wheat dough with varying amounts of hazelnut or walnut oil cakes. However, it could improve in directly addressing the research aims outlined in the introduction. While the data are comprehensive, linking each specific result to the original research objectives would enhance coherence.
Discussion: The discussion section lacks depth in relating the obtained results to existing theories or prior studies. Strengthening this section by integrating the results into the existing scientific literature could significantly enhance its impact.
Conclusion: The conclusion effectively summarizes the main outcomes of the study. However, it could be more robust by explicitly discussing the implications of the findings within the context of the study's hypothesis and potential future research directions.
Overall Evaluation:
While the manuscript offers valuable experimental data, it lacks a clearly articulated hypothesis or research question.
The study's objectives and hypothesis could be better aligned for a more coherent narrative and to support the interpretation of results.
Suggested Improvements:
Hypothesis Clarity: Establish a clear hypothesis in the introduction, explaining the expected impact of substituting wheat flour with hazelnut or walnut oil cakes on rheological properties.
Coherence in Sections: Ensure a stronger alignment between the introduction, methodology, results, and discussion to maintain a coherent narrative.
Comprehensive Discussion: Strengthen the discussion by relating the results to existing literature and discussing their broader implications within the context of the established hypothesis.
Revised Conclusion: Revise the conclusion to emphasize the significance of the findings in relation to the study's hypothesis and possible avenues for future research.
By addressing these points, the manuscript's clarity, coherence, and contribution to the field would be significantly improved.
Author Response
Reviewer 5
Title: The title effectively conveys the focus of the study: analyzing the rheological properties of wheat dough with hazelnut or walnut oil cakes. It accurately reflects the main components and scope of the research.
Response: Thank you for this review
Abstract: The abstract provides a concise yet informative overview of the study. However, it could benefit from a more explicit articulation of the hypothesis or specific research question guiding the study.
Response: the abstract section was checked and improved in accordance with the suggestions of other reviewers. The hypothesis was adedd.
Introduction: The introduction offers a comprehensive background on dough rheology and the potential impact of adding hazelnut or walnut oil cakes. The introduction offers contextual information but lacks a clearly defined research problem or a well-stated hypothesis. A stronger articulation of the research problem, along with a well-defined hypothesis, would greatly enhance the manuscript's foundation.
Response: In the literature review, we focused on the influence of equal ingredients and enrichment additives on the rheological characteristics of wheat flour-based dough. In our previous work, we analyzed the impact of ground nuts on dough characteristics, as such ingredients are attractive in terms of enriching and improving the nutritional value of bakery products. An equally interesting ingredient for cakes and bakery products may be ground nut cakes as a by-product after oil pressing. Not only are they a rich source of protein, fiber and minerals or other biologically active substances of plant origin, but their use fits well into the general trend of waste-free production and maximum use of by-products. Our proposed research hypothesis indicates that replacing part of wheat flour with ground nut cake modifies the rheological characteristics of the dough and is closely related to the purpose of the work.
Materials and Methods: This section provides detailed information about the materials and methodologies employed. However, it lacks explicit connection to the study's aims or hypotheses. Consider organizing the section into subsections based on the experimental procedures. Create clear headings for different experimental techniques or analyses conducted (e.g., Farinograph Analysis, Extensograph Analysis, Oscillatory Rheometer Testing).
Response: The methods section has been supplemented and arranged in accordance with the suggestions of other reviewers
Results: The results section presents extensive data and analyses related to the rheological properties of wheat dough with varying amounts of hazelnut or walnut oil cakes. However, it could improve in directly addressing the research aims outlined in the introduction. While the data are comprehensive, linking each specific result to the original research objectives would enhance coherence.
Discussion: The discussion section lacks depth in relating the obtained results to existing theories or prior studies. Strengthening this section by integrating the results into the existing scientific literature could significantly enhance its impact.
Conclusion: The conclusion effectively summarizes the main outcomes of the study. However, it could be more robust by explicitly discussing the implications of the findings within the context of the study's hypothesis and potential future research directions.
Overall Evaluation:
While the manuscript offers valuable experimental data, it lacks a clearly articulated hypothesis or research question.
Respnse: the hypothesis was articulated in introduction section
The study's objectives and hypothesis could be better aligned for a more coherent narrative and to support the interpretation of results.
Response: the manuscript was corrected and improved
Suggested Improvements:
Hypothesis Clarity: Establish a clear hypothesis in the introduction, explaining the expected impact of substituting wheat flour with hazelnut or walnut oil cakes on rheological properties.
Response: the hypothesis was added to the intrduction section
Coherence in Sections: Ensure a stronger alignment between the introduction, methodology, results, and discussion to maintain a coherent narrative.
Response: the manuscript was corrected and improved.
Comprehensive Discussion: Strengthen the discussion by relating the results to existing literature and discussing their broader implications within the context of the established hypothesis.
Response: The discussion in the paper has been significantly improved according to the reviewers' suggestions. The research results were compared with the observations of other researchers dealing with dough rheology.
Revised Conclusion: Revise the conclusion to emphasize the significance of the findings in relation to the study's hypothesis and possible avenues for future research.
Response: the conclusions section was improved
By addressing these points, the manuscript's clarity, coherence, and contribution to the field would be significantly improved.
Response: The entire manuscript has been improved. Individual sections have been supplemented according to the Reviewers' tips and suggestions. We hope that this form is correct.

Round 2
Reviewer 1 Report
Comments and Suggestions for Authors
Comments: Although the authors carefully revised this manuscript, and added the data of composition in hazelnut and walnut oil cakes powder, they cited their recent paper (Pycia, K.; Juszczak, L. Influence of hazelnut and walnut oil cakes powder on thermal and rheological properties of wheat flour. Foods 2023, 12, 4060. https:// doi.org/10.3390/foods12224060) to show the contents of ash, fat and fiber after the wheat flours mixtures with hazelnut or walnut oil cakes. I still want to know why they added the hazelnut (HOC) or walnut oil cakes (WOC) powder at the level of 5%, 10% and 15%? What is the basis? HOC has a protein content of 36.4%, WOC has a protein content of 42.7%. After they separately mix with wheat flours, the contents of protein, fat, fiber and ash in the flour mixture should be given this manuscript, because these parameters have limit levels in common wheat flour commodities. Thus they can explain why they add HOC or WOC powder at levels of 5%, 10%, or 15%, they can also explain the difference in rheological characteristics for flour mixtures, and which addition is better for use. They can also indicate the further study direction.
In addition, the data is commonly show Mean±SDa, but Meana±SD is rare.
Author Response
Reviewer 1
Comments: Although the authors carefully revised this manuscript, and added the data of composition in hazelnut and walnut oil cakes powder, they cited their recent paper (Pycia, K.; Juszczak, L. Influence of hazelnut and walnut oil cakes powder on thermal and rheological properties of wheat flour. Foods 2023, 12, 4060. https:// doi.org/10.3390/foods12224060) to show the contents of ash, fat and fiber after the wheat flours mixtures with hazelnut or walnut oil cakes. I still want to know why they added the hazelnut (HOC) or walnut oil cakes (WOC) powder at the level of 5%, 10% and 15%? What is the basis? HOC has a protein content of 36.4%, WOC has a protein content of 42.7%. After they separately mix with wheat flours, the contents of protein, fat, fiber and ash in the flour mixture should be given this manuscript, because these parameters have limit levels in common wheat flour commodities. Thus they can explain why they add HOC or WOC powder at levels of 5%, 10%, or 15%, they can also explain the difference in rheological characteristics for flour mixtures, and which addition is better for use. They can also indicate the further study direction.
Response: The research results presented in the work are a continuation of the research topic related to the analysis of the influence of hazelnuts or walnuts, as well as by-products in the form of oil cakes, on the thermal and rheological properties of wheat flour and the rheological properties of dough. In all works (molecules-26133969, materials-1545382, foods-2674277) the addition of nuts or nut oil cake was used at levels of 5, 10 and 15%. These levels have been selected experimentally so that the difference between mixtures can be noticed. Too low an addition could have a statistically insignificant impact on the tested parameters. Moreover, too low a share of oil cake as an enriching ingredient will not increase the nutritional value of the final product, while too much addition may result in obtaining dough as a semi-finished product that will not be suitable for baking bread. The analysis of the literature on the subject also indicates that researchers analyzed the impact of various levels, including: cashew nut proteins (5%, 10%, 15%, 20%) (Azzez et al. 2021), chestnut flour (20%, 40%, 60%) (Dokic et al. 2014), pistachio cake (5% , 15%, 25%) (Guardianelli et al. 2023) or bran at the level of 5-25% (Li et al. 2023) on the rheological properties of the dough. In our opinion, the levels tested are appropriate and enable observation of the direction of changes taking place. Accorging due to Reviewer’s suggestion in the research material section, information regarding moisture content, the content of protein, fat, fiber and minerals in flour mixtures containing HOC and WOC has been added. In our opinion, further research results in which bread is planned to be baked based on mixtures containing ground cakes and its characteristics, including sensory characteristics, may be decisive in determining which of the ingredients used will be better. Research conducted in this way will allow obtaining the opinions of potential consumers regarding the final preferences of such enriched products.
In addition, the data is commonly show Mean±SDa, but Meana±SD is rare.
Response: Thank you for your attention. The data in the tables has been corrected according to the Reviewer's suggestions.

Reviewer 2 Report
Comments and Suggestions for Authors
Accept
Comments on the Quality of English Language
Accept
Author Response
Reviewer 2
Accept
Response: The authors of this paper would like to thank you very much for this opinion.

Reviewer 3 Report
Comments and Suggestions for Authors
The authors performed the points raised by the reviewer. However, there are still issues to be addressed.
1. Lines 30, G' and G" should be referred to as storage and loss moduli.
2. Lines 49-50, for bread strong flours are recommended. Light flours, which have lower protein content is more suitable for cakes. Also, "health point of view" is not clear. Removal of this statement is suggested. As the following two statements are good enough for justification of the study.
3. Please change "research material" to "materials"
4. Instead of "rheometric analyses", the use of "Rheological measurements" is suggested
5. Line 125: Instead of "G' module", use "storage modulus" or G', same for the loss modulus.
6. Line 126 and rest of the article, "frequency sweep test" should be used instead of "sweep frequency test".
7. All the results should be given with their Error values, for example, line 186 (52.3 ±)
8. Line 204-205; Rewrite the sentence for clarity.
9. Line 209: Please correct- "affected" not "effected".
10. Line 225: Please avoid using "you", "we" in the text.
11. Line 258-260: Rewrite the sentence as follows if the authors mean:
Jakubczyk and Haber [1723] and Achremowicz et al. [1824] used DDT to determine the quantity and quality of gluten in flour and its water absorption capacity.
12. Please correct the first sentence of the conclusion
Remove "The analyses of the test " and start as " The results indicate that ....."
13. The conclusion needs to be rewritten to be more to the point, avoid duplicated phrases in the same sentence such as "the use of" in the lines 694-695.
Comments on the Quality of English Language
It needs to be double checked, several grammatical errors and long sentences need to be corrected.
Author Response
Reviewer 3
The authors performed the points raised by the reviewer. However, there are still issues to be addressed.
- Lines 30, G' and G" should be referred to as storage and loss moduli.
Response: the sentence has been corrected
- Lines 49-50, for bread strong flours are recommended. Light flours, which have lower protein content is more suitable for cakes. Also, "health point of view" is not clear. Removal of this statement is suggested. As the following two statements are good enough for justification of the study.
Response: the sentence has been corrected
- Please change "research material" to "materials"
Response: the sentence has been corrected
- Instead of "rheometric analyses", the use of "Rheological measurements" is suggested
Response: the sentence has been corrected
- Line 125: Instead of "G' module", use "storage modulus" or G', same for the loss modulus.
Response: the sentence has been corrected
- Line 126 and rest of the article, "frequency sweep test" should be used instead of "sweep frequency test".
Response: the sentence has been corrected
- All the results should be given with their Error values, for example, line 186 (52.3 ±)
Answer: Thank you for this suggestion. The tables provide average values of individual parameters along with standard deviations. In the text, when parameter values are given for individual samples, SD is also provided, according to the Reviewer's suggestion. Furthermore, when discussing results, authors often provide an average value for a group of samples. In this case, the standard deviation is not calculated.
- Line 204-205; Rewrite the sentence for clarity.
Response: the sentence has been corrected
- Line 209: Please correct- "affected" not "effected".
Response: the sentence has been corrected
- Line 225: Please avoid using "you", "we" in the text.
Response: the sentence has been corrected
- Line 258-260: Rewrite the sentence as follows if the authors mean:
Jakubczyk and Haber [1723] and Achremowicz et al. [1824] used DDT to determine the quantity and quality of gluten in flour and its water absorption capacity.
Response: the sentence has been corrected
- Please correct the first sentence of the conclusion
Remove "The analyses of the test " and start as " The results indicate that ....."
Response: the sentence has been corrected
- The conclusion needs to be rewritten to be more to the point, avoid duplicated phrases in the same sentence such as "the use of" in the lines 694-695.
Response: the sentence has been corrected

Reviewer 5 Report
Comments and Suggestions for Authors
Authors have thoroughly addressed all comments and suggestions, significantly improving the manuscript.
Author Response
Reviewer 5
Authors have thoroughly addressed all comments and suggestions, significantly improving the manuscript.
Response: The authors of this paper would like to thank you very much for this opinion.
